# Amphetamine-like Deferiprone and Clioquinol Derivatives as Iron Chelating Agents

**DOI:** 10.3390/molecules29174213

**Published:** 2024-09-05

**Authors:** Mahmoud El Safadi, Katie A. Wilson, Indigo J. Strudwicke, Megan L. O’Mara, Mohan Bhadbhade, Tristan Rawling, Andrew M. McDonagh

**Affiliations:** 1School of Mathematical and Physical Sciences, Faculty of Science, University of Technology Sydney, Sydney, NSW 2007, Australia; mahmoudelsafadi@uaeu.ac.ae (M.E.S.); tristan.rawling@uts.edu.au (T.R.); 2Department of Chemistry, College of Science, United Arab Emirates University, Al Ain P.O. Box 15551, United Arab Emirates; 3Research School of Chemistry, The Australian National University, Canberra, ACT 2601, Australia; k.wilson@mun.ca (K.A.W.); indigo.strudwicke@anu.edu.au (I.J.S.); m.omara@uq.edu.au (M.L.O.); 4Department of Biochemistry, Memorial University of Newfoundland, St. John’s, NL A1C 5S7, Canada; 5Australian Institute for Bioengineering and Nanotechnology, The University of Queensland, St. Lucia, QLD 4067, Australia; 6Mark Wainwright Analytical Centre, The University of New South Wales, Sydney, NSW 2052, Australia; m.bhadbhade@unsw.edu.au

**Keywords:** iron chelator, amphetamine-type chelator, antioxidant, molecular dynamics, human dopamine transporter

## Abstract

The accumulation of iron in dopaminergic neurons can cause oxidative stress and dopaminergic neuron degeneration. Iron chelation therapy may reduce dopaminergic neurodegeneration, but chelators should be targeted towards dopaminergic cells. In this work, two series of compounds based on 8-hydroxyquinoline and deferiprone, iron chelators that have amphetamine-like structures, have been designed, synthesized and characterized. Each of these compounds chelated iron ions in aqueous solution. The hydroxyquinoline-based compounds exhibited stronger iron-binding constants than those of the deferiprone derivatives. The hydroxyquinoline-based compounds also exhibited greater free radical scavenging activities compared to the deferiprone derivatives. Molecular dynamics simulations showed that the hydroxyquinoline-based compounds generally bound well within human dopamine transporter cavities. Thus, these compounds are excellent candidates for future exploration as drugs against diseases that are affected by iron-induced dopaminergic neuron damage, such as Parkinson’s disease.

## 1. Introduction

The accumulation of iron in the human brain has been implicated in neurodegenerative diseases such as Parkinson’s disease [1]. Iron accumulation can play a role in oxidizing endogenous compounds into cytotoxic species that cause oxidative stress and progressive neuron degeneration [2,3,4]. In particular, the accumulation of iron within the substantia nigra may be an important contributor to the progression of Parkinson’s disease [5]. Within dopaminergic neural cells, dopamine can be oxidized into toxic quinone-type species [6], and excess iron can facilitate such oxidative processes [7], as well as others [8], to promote dopaminergic cell death.

Iron chelation therapy has been proposed to reduce dopaminergic neurodegeneration [2,5,9] by preferentially binding to iron [10]. Deferiprone and clioquinol (Figure 1) have been investigated extensively in this regard [2,5,10,11,12,13,14,15]. They have high selectivities toward transition metals, are orally active, and can cross the blood-brain barrier [11,16,17]. There is, however, concern that the removal of iron from regions of the brain may produce adverse side effects, as iron is essential to normal brain function [12,18]. To gain further understanding of these complex processes, it would be beneficial to target the region of the brain where dopaminergic neurons are most abundant. Amphetamine-type compounds can fulfil this role. Amphetamine-like compounds can enter presynaptic neurons through dopamine transporter (DAT) proteins and compete for reuptake with dopamine. DAT is selectively expressed in dopaminergic cells, and given that the substantia nigra is rich in these cells, it is plausible that an amphetamine-like compound with the capacity to bind iron(III) might target and transport it from these areas. Recently, deferiprone-like molecules were synthesized that contain alanine groups (thus resembling L-DOPA), exhibited good iron(III) chelating ability and were protective to neuronal cells [19]. Brain-permeable compounds based on 8-hydroxyquinoline have also shown excellent neuroprotective effects [20,21,22].

Thus, the aim of the current work is to investigate iron chelators that have the potential to preferentially target dopaminergic neurons. Here, we explore the design and synthesis of compounds modelled on dopamine reuptake inhibitors that also possess the ability to chelate iron, focusing on amphetamine-like structures in particular.

## 2. Results and Discussion

### 2.1. Design and Synthesis of the Target Compounds

In the designed target structures, the aromatic ring of amphetamine (Figure 1) was replaced with either 3-hydroxy-2-methylpyridinone or 8-hydroxyquinoline groups, which are the iron-chelating regions of deferiprone and clioquinol, respectively. Iron in the central nervous system exists in the iron(II) or ferrous state as well as in the iron(III) or ferric state [22]. Here, we chose 3-hydroxy-4-pyridinone and 8-hydroxyquinoline groups as chelating moieties, as both have been previously shown to have a strong affinity to iron(III) [21,23,24].

In the deferiprone-based compounds **1**–**5**, the aminopropyl side chain was covalently attached to the heterocyclic nitrogen atom. Analogues were prepared as the racemate (**1**), and the R- and S-enantiomers (compounds **2** and **3**, respectively). Deferiprone analogues **4** and **5** were included to assess the effects of modifying the aminopropyl side chain on iron-binding properties. Compounds **6**–**9** were 8-hydroxyquinoline-based compounds with the aminopropyl group located in different positions around the 8-hydroxyquinoline ring, and **10** possessed no nitrogen atom in the side chain.

The deferiprone-based compounds were synthesized by first protecting the hydroxyl group of maltol using benzyl bromide (Figure 1) to avoid subsequent interactions with amines. The resultant benzyloxy-2-methyl-4H-pyran-4-one was reacted with the appropriate alkyl amine compounds, followed by deprotection of the benzyl group to produce compounds **1**–**5**. Single-crystal X-ray crystallography studies of compounds **3**–**5** (see Appendix A) showed the expected structures (note that deferiprone-based compounds can be protonated/deprotonated depending on the pH [25]). Thus, compounds **3** and **4** were crystallized as dihydrochlorides, and compound **5** was crystallized as the hydrochloride.

8-Hydroxyquinoline compounds (Figure 2) were prepared by the reaction of 8-hydroxyquinoline bearing a carbaldehyde group (in either the 2, 4, 5 or 7 positions) [26,27,28] with nitroethane [29] to form the corresponding nitropropenyl compounds, which were subsequently reduced to produce the corresponding amine compounds (**6**–**9**). Compound **10** was prepared from 2-methylquinolin-8-ol, whereby the hydroxy group was protected using iodomethane to give the corresponding 8-methoxy compound. An isopropyl moiety was added to the 2-position using a published procedure [30], followed by demethylation to produce compound **10**.

### 2.2. Iron Chelation

The capacity of the new compounds to chelate iron was examined using mass spectrometry, UV-visible spectroscopy and isothermal titration calorimetry (ITC). Solutions of **1**–**9** and Fe(NO_3_)_3_ with ligand–iron(III) ratios from 0.5:1 to 6:1 were mixed and then examined using UV-visible spectroscopy. Figure 2 shows a representative set of spectra obtained using compound **1**. A peak at ~460 nm was assigned to metal-ligand charge transfer (MLCT). The wavelength of this peak is similar to that used to monitor formation of the Fe^III^(deferiprone)_3_ complex under similar conditions [10]. The intensity of the MLCT band increased by only a very small amount once a ligand ratio of 3:1 (blue curve in Figure 2) was obtained, indicating that the iron was fully coordinated with three ligands. As the ligand–iron(III) ratio increased past 3:1, the absorption band at 300 nm increased in intensity and shifted to ~280 nm, close to that of free ligands, indicating that unbound ligands remained in the solution after the 3:1 ratio was obtained. This suggests that the iron was fully co-ordinated at the 3:1 stoichiometry. Compounds **2**–**5** displayed similar behaviour to that of compound **1** under the same conditions (see Appendix A). We synthesized [Fe(**L**)_3_] separately, where L = **1**–**5** (see Appendix A), and determined the molar absorptivities of the MLCT bands to be in the range of ~4200–~4300 M^−1^cm^−1^, which are similar to the published value for [Fe^III^(deferiprone)_3_] of 4600 M^−1^cm^−1^ [10]. Compounds **6**–**9** also displayed similar coordinating properties, i.e. the formation of [Fe(**L**)_3_] complexes for L = **6**–**9**, but with peaks assigned to MLCT transitions at ~590 nm and ~460 nm.

The mass-spectra of the deferiprone-based complexes (see Appendix A) contained peaks assigned to FeL_3_, where L = the singly deprotonated compounds, as well as some FeL_2_, which presumably formed during the MS ionization process, although we cannot rule out small amounts being present from the synthesis. The mass-spectra of the iron(III) complexes with 8-hydroxyquinoline-based compounds **6**–**9** also contained major peaks assigned to FeL_3_, as well as some assigned to FeL_2_.

Table 1 shows isothermal titration calorimetry (ITC) data that describe solution thermodynamic interaction parameters such as stoichiometry of the interaction (n), the association constant (K_a_), the free energy (ΔG), enthalpy (ΔH) and entropy (ΔS). Titration with Fe(NO_3_)_3_ revealed a stoichiometric ratio of 0.33 ± 0.1 for compounds **1**–**5** and 2.8 ± 0.2 for compounds **6**–**9**. These ratios were consistent with the mass spectrometry and UV-visible spectroscopy data and confirmed the FeL_3_ complex’s stoichiometry. Each of the iron(III) complexes paired with complexes **1**–**9** had ΔH^0^ < 0, indicating an exothermic reaction upon complexation. In each case, |ΔH^0^| > |TΔS^0^| and ΔG^0^ < 0; thus, the reactions were spontaneous and enthalpy-driven. The binding constants, K_a_, for compounds **1**–**5** with iron(III) were ~2–3 × 10^4^, and for compounds **6**–**9**, they were ~1–2 × 10^5^. Thus, the 8-hydroxyquinoline-based compounds were stronger iron chelators than the deferiprone-based compounds. 

### 2.3. Anti-Oxidant Activity

Oxidative damage to dopaminergic neurons is implicated in neurodegenerative diseases such as Parkinson’s disease, so therapeutic agents with antioxidant activity may offer protection. The antioxidant activities of compounds **1**–**9** were evaluated using a 1,1-diphenyl-2-picrylhydrazyl (DPPH) scavenging assay {Sharma, 2009 #104} using ascorbic acid as reference. DPPH (Figure 3) is a stable radical that can be scavenged by antioxidants. The radical scavenging activities of compounds **1**–**9** are shown in Figure 3 and Table 2.

Compounds **1**–**5** showed relatively low radical scavenging activities. The stabilization of phenoxyl radicals by strong hydrogen bonding with an adjacent −OH group and the lack of extended conjugation in compounds **1**–**5** were reflected in their low radical scavenging ability, consistent with previous work investigating the antioxidant activity of deferiprone [31] where only weak antioxidant activity was observed. The 8-hydroxyquinoline-based compounds **6**–**9** exhibited superior radical scavenging activity compared to that of compounds **1**–**5**. Unsubstituted 8-hydroxyquinoline has been shown to exhibit good antioxidant behavior, even in the presence of iron [32]. The greatest antioxidant activity within the 8-hydroxyquinoline-based compounds was observed for **6** and **8**, which have the propylamine group *ortho* to either the pyridyl N atom (**6**) or the OH group (**8**). Previous work has shown that electron-donating groups at the 2-position of 8-hydroxyquinoline increase antioxidant activity [33]. Overall, these trends (compared to the reference material ascorbic acid) were consistent with those that might be expected by examination of the molecular structures. Ascorbic acid possesses two oxidizable OH functional groups that can donate two protons and thus effectively scavenge the DPPH free radical. On the other hand, compounds **1**–**9** may donate single protons from the phenolic OH groups, although in some cases, the nearby extensive conjugation makes this quite a favourable process.

### 2.4. Molecular Dynamics Simulations

The interaction of the compounds within the DAT-binding cavity were examined using molecular dynamics (MD) simulations. The simulations were analysed for contact frequency, which indicated the percentage of the simulation time that residues in the binding site were within a distance of 4 Å from the substrate. Dopamine was used as the positive control, given its biogenic interaction with DAT. As expected, dopamine was shown to be very stable within its binding cavity in hDAT, with a root-mean-square deviation (RMSD) of 2.8 ± 0.6 Å (Appendix A). Dopamine had contact with nine residues for >90% of the total simulation time (Figure 4A and Appendix A). The most frequent of these was Ser149, which formed a hydrogen bond with dopamine, followed by Tyr156, which interacted with the aromatic ring of dopamine through a parallel π-stacking interaction (Figure 4B). A sodium ion also played an important role in the binding cavity and was almost always coordinated to the amine group on dopamine.

Simulations showed that 8-hydroxyquinoline-based compounds **6**–**9** generally bound well within the hDAT binding cavity. Both isomers of compound **8** bound well within the cavity of hDAT, though the R-enantiomer (***R*-8**) had a lower RMSD at 3.9 ± 0.9 Å compared with the S-enantiomer (***S*-8**, 5.4 ± 1.2 Å, Appendix A). ***R*-8** was in contact with 10 residues for >90% of the total simulation time and an additional three residues for >80% of the total simulation time (Figure 4A). The most frequent interactions for ***R*-8** were with Ser149 and Tyr156. ***S*-8** also formed many interactions in the binding cavity (eight residues in contact for > 90% of the simulation, Figure 4A). ***R*-8** coordinated to the sodium ion in hDAT for 85% of the total simulation time, compared to 32% in ***S***-**8** (Appendix A).

Compared to compound **8**, compounds **6** and **7** bound less tightly to hDAT. The R- and S-enantiomers of compound **7** (***R*-7** and ***S*-7**) were bound with approximately even stability, with RMSD values of 4.4 ± 1.2 Å and 4.1 ± 1.1 Å, respectively (Appendix A). While both enantiomers remained bound in the general region of the dopamine binding site, the position of each substrate throughout the simulation with respect to the binding cavity was seen to change considerably throughout the simulation. ***R*-7** had contact with seven residues for more than 90% of the total simulation, while ***S*-7** interacted with three residues for more than 90% of the total simulation (Figure 4A). Compound ***R*-6** held well in the binding cavity, with a comparatively minimal change of orientation relative to its starting position, the lowest RMSD value and least relative movement observed for all non-dopamine substrates examined (RMSD = 3.6 ± 0.7 Å; Appendix A). Similar to dopamine, the amine of ***R*-6** was coordinated to a sodium ion frequently at 93% of the total simulation time and was in proximity to seven protein residues for more than 90% of the total simulation (Figure 4A), and an additional five for over 80% of the total simulation (Figure 4A). The *S*-enantiomer ***S*-6** bound a sodium atom for 56% of the simulation and had contact with just four residues for more than 90% of the simulation (Appendix A). As expected, the non-amphetamine **10** did not coordinate a sodium ion, and although it did have some interaction with hDAT, the binding was very unstable with an RMSD of 5.3 ± 2.4 Å (Appendix A).

All of the deferiprone analogues (**2**–**5**) moved from the initial docked position in the dopamine binding site, leading to large RMSD values (up to 5.4 Å, Appendix A). The largest movement was observed for compound **3**, which moved into the main channel of hDAT. Due to the instability in their binding locations, compounds **2**, **3** and **5** had more than three contacts for >90% of the total simulation time. Compound **4** formed four contacts for >90% of the simulation. None of the deferiprone deviates coordinated a sodium ion (Appendix A). The instability of the deferiprone derivatives in the hDAT binding cavity suggests that these compounds will not act as DAT substrates.

## 3. Materials and Methods

The syntheses of **1**–**10** and associated precursors are described in the Appendix A. The UV-visible spectra were recorded using an Agilent Technologies Cary 60 spectrometer (Santa Clara, CA, USA). The infrared spectra were collected using a ThermoScientific FT-IR (ATR) Nicolet 6700 spectrometer in the range of 4000–600 cm^−1^. Isothermal titration calorimetry (ITC) experiments were carried out at 25 °C on a Nano-ITC instrument (TA instrument) with 300 rpm. All of the compounds and iron were dissolved in 10% MeOH in PBS with pH = 7.4. In the cases of **1**–**5**:Fe complexes, **1**–**5** (0.3 mM, 170 µL) were placed in a sample cell, and Fe(NO_3_)_3_·9H_2_O (3 mM, 50 µL) was placed in a syringe, followed by the addition of 1.9 µL from a burette to the cell every 100 s. In the case of the **6**–**9** complexes, **6**–**9** (3 mM, 50 µL) were placed in a syringe, and Fe(NO_3_)_3_·9H_2_O (0.15 mM, 150 µL) was placed in a sample cell, followed by the addition of 1.9 µL of complexes **6**–**9** from the burette to the cell every 100 s. Titration was performed in a sequence of 25 injections in both methods. The raw data were processed using Nano analyze software.

Antioxidant activity was measured using a 1,1-diphenyl-2-picrylhydrazyl (DPPH) assay performed according to a modified published method [34]. Stock solutions (10 mM in methanol) of **1**–**9** and a 200 μM methanolic DPPH solution were prepared. A stock solution of ascorbic acid (1 mM in methanol) was used as a control. DPPH (200 μΜ, 50 μL) was added to 150 μL of sample solutions (ascorbic acid, **1**–**9**) with different concentrations using 96 well microplates. Each of the additions was performed in triplicate. The microplates were wrapped in aluminium foil and kept at 30 °C for 30 min in the dark. Spectrophotometric measurements were then recorded using a Thermolabsystems Multiskan Ascent spectrophotometer at 520 nm. The measurements were performed while ensuring no exposure to ambient light. Activities were calculated as follows:DPPH radical scavenging activity (%) = [(Abs_control_ − Abs_compound_)]/(Abs_control_)] × 100
where Abs_control_ is the absorption of (200 μM, 50 μL) DPPH with 150 μL of neat methanol, and Abs_compound_ is the absorption of test compounds at different concentrations plus DPPH (200 μM, 50 μL).

The modelling and pharmacokinetic parameters were calculated using Discovery Studio 4.5 (Accelrys, San Diego, CA, USA). Systems were simulated using methods similar to previous work done by the O’Mara group [35]. A homology model of hDAT was built using the PHYRE2 server [36], with dDAT as the template structure (PDB ID: 4M48 [37]). Each compound was docked into the dopamine binding site of the proteins using AutoDock Vina [38] incorporating a flexible docking approach. During MD simulations, the protein and lipids were modeled with the GROMOS54a7 forcefield [39], while parameters for each compound were generated using the Automated Topology Builder [40]. hDAT was embedded in a membrane containing 80% POPC and 20% cholesterol. The systems were solvated with SCP water and 0.15 NaCl that contained additional counter ions to neutralize the system. Each system was then minimized using the steepest descent algorithm and equilibrated with decreasing constraints on the protein (1000, 100, 500, 50 and 10 kJ mol^−1^ nm^−2^) through five 1 ns simulations. Triplicate 500 ns molecular dynamics simulations were run using a 2 fs timestep with the periodic boundary conditions. All simulations were run using Gromacs 2019.4 [39,41]. Throughout all simulations, a constant pressure (1 bar semi-isotropic position scaling using the Berendsen barostat, with coupling constant τ_P_ = 0.5 ps and isothermal compressibility = 4.5 × 10^−5^ bar) and temperature (300 K using the Bussi-Donadio-Parrinello velocity rescale thermostat, coupling constant τ_T_ = 0.1 ps) were maintained. The triplicate set for each compound was combined into one simulation of 1500 ns, and once prepared, these simulations were analyzed in Visual Molecular Dynamics (VMD) [42] to assess the binding modes within the cavity. The simulations were analysed for substrate RMSD and contact frequency, which indicated the percentage of the simulation time that residues in the binding site were within 4 Å of the substrate.

## 4. Conclusions

Compounds with amphetamine-like structures based on deferiprone and 8-hydroxyquinoline were designed, synthesized and characterized. These compounds were designed to chelate iron, which has been implicated in the progression of Parkinson’s disease. We showed that each of the compounds **1**–**10** coordinated iron in aqueous solution. Single-crystal X-ray studies revealed the structures of the synthesised compounds. Isothermal calorimetry, mass spectrometry, and UV-visible spectroscopy studies showed that the compounds chelated iron in a 3:1 ratio, and the 8-hydroxyquinoline-based compounds had stronger binding constants than those of the deferiprone derivatives. In terms of antioxidant properties, the 8-hydroxyquinoline-based compounds exhibited better free radical scavenging activities than that of the deferiprone derivatives, although none were greater than Vitamin C.

Molecular modelling of the interaction of the new compounds with DAT indicated that several compounds (**6**–**8**) have similar binding properties to those of dopamine. Thus, it is possible that these compounds may be effectively transported within dopaminergic cells and chelate iron, as well as reduce iron-induced oxidative stress. Further work will be required to explore whether the excellent DAT binding properties of selected compounds and any subsequent transportability will affect their ability to chelate and transport iron.

## Data Availability

CCDC 2204668, 2204672, 2204673, 2214096, 2214097, and 2214098 contain the supplementary crystallographic data for this paper. These data can be obtained free of charge via www.ccdc.cam.ac.uk/data_request/cif (26 August 2024). Computational data is available from https://github.com/OMaraLab/Amphetamine-like_iron_chelators (26 August 2024). Computational data is available from https://doi.org/10.5281/zenodo.13645793 (26 August 2024).

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
