# Peer review of "Amphetamine-like Deferiprone and Clioquinol Derivatives as Iron Chelating Agents"

_molecules, 2024, doi:10.3390/molecules29174213_

Round 1

Reviewer 1 Report

Comments and Suggestions for Authors

This manuscript reports on the synthesis of two families of compounds, containing either 3-hydroxy-2-methylpyridinone or 8-hydroxyquinoline, which upon deprotonation behave as metal ion ligands. These compounds incorporate the typical motif of amphetamine, so as to be able to enter the presynaptic neurons. The aim of the study is to develop compounds able to selectively remove iron from dopaminergic cells and act as dopamine antioxidants. This work is interesting for the readership of Molecules. The objectives are clear, and the experiments clearly indicate that the family of synthetic molecules derived from clioquinol are better at performing the desired functions than those derived from deferriprone. I fully support publication of this nice work in Molecules without any change.

Author Response

Comment 1: I fully support publication of this nice work in Molecules without any change.

Response 1: We thank the reviewer for their consideration of our manuscript.

Reviewer 2 Report

Comments and Suggestions for Authors

Dear Authors,

I have read your manuscript “Amphetamine-like Deferiprone and Clioquinol Derivatives as Iron Chelating Agents” and I think the topic could be of interest to the scientific community, however, I found major issues that need to be addressed to improve the quality of your work, before considering its publication.

-          Introduction: it should be enlarged especially in reporting recent results in the use of iron sequestering agents. Furthermore, the authors should report the metal oxidation state they are targeting and the characteristics of a selective Fe(III) chelator.

-          The syntheses are reported as supplementary information, please cite this in the “Materials and Methods” section. Concerning the syntheses, the elemental analysis results are not stated. These analyses are important to confirm the purity of the compounds and predict the correct chemical formula that has to be added. Most of the compounds are pictured with protonated amine, so they are probably isolated in the hydrochloride form. Did the Authors consider it for further investigations and solutions’ concentration?

-          All the ligands are weak acids/bases, so it is of pillar importance to know their speciation in physiological conditions. Are they protonated? Neutral? This is also a key point in predicting ADME and biodistribution. Additional measurements to estimate acid dissociation constants would improve the manuscript.

-          The iron complexation study is extremely poor and not enough to support the aim reported in the title. A thorough investigation should be carried out to estimate the stability constants of the metal complexes and their speciation as a function of pH, so to predict the pFe3+ (-log[Fe3+free]) and compare the results with those of the leading compounds (deferiprone and Clioquinol). The Authors need to consider iron-hydroxo species that may form in solution as well. Furthermore, the Authors only report the data on compounds 1 and 6 (as supplementary), the main differences in absorption bands and at least a picture with Fe(III) complexes at the same molar ratio with all the ligands has to be added and commented on.

-          Antioxidant activity with DPPH assay: compound 10 is missing. EC50 (the antioxidant concentration necessary to decrease the initial amount of DPPH radical by 50%) could help compare the results. I am concerned about the methods the Authors used to assess antiradical activity. To my knowledge, in this assay, the absorbance is read at the maximum of DPPH (517 nm) and the decrease is used to define the EC50 as profusely reported (an example   https://doi.org/10.1111/cbdd.12847)

-          The Authors obtained the solid complexes and carried out some IR measurements. Are these data useful to the scope of the manuscript? Were the isolated complexes used to confirm the stoichiometry predicted in solution? If these results can provide some information, they should be properly added to the manuscript throughout the discussion. It would be interesting to test the isolated iron complexes for radical scavenging ability and compare them with the free ligands.

-          The high affinity for the dopamine transporter could be on one side, as stated in the rationale of the manuscript, a good way of targeting the brain, but on the other side, it could be detrimental to iron complexation, being the transporter in competition to bind the chelator itself. The Authors should comment on this in the discussion. 

Comments on the Quality of English Language

The manuscript is readable and the quality of English is fine.

Author Response

Comment 1: Introduction: it should be enlarged especially in reporting recent results in the use of iron sequestering agents.

Response 1: Addional text and references have been included as requested.

Comment 2: … the authors should report the metal oxidation state they are targeting and the characteristics of a selective Fe(III) chelator.

Response 2: This has been included as requested.

Comment 3: The syntheses are reported as supplementary information, please cite this in the “Materials and Methods” section.

Response 3: This has been done as requested.

Comment 4: Concerning the syntheses, the elemental analysis results are not stated. These analyses are important to confirm the purity of the compounds and predict the correct chemical formula that has to be added. Most of the compounds are pictured with protonated amine, so they are probably isolated in the hydrochloride form. Did the Authors consider it for further investigations and solutions’ concentration?

Response 4: Yes, the protonation state of the compounds was carefully considered and all final compounds were reported as the di- or hydrochloride salts (supplementary information) and the appropriate molecular weights were used when preparing drug stocks. We have included copies of NMR spectra to support compound purity and structures were robustly characterised by 13C and 1H NMR, HRMS and IR.

Comment 5: All the ligands are weak acids/bases, so it is of pillar importance to know their speciation in physiological conditions. Are they protonated? Neutral? This is also a key point in predicting ADME and biodistribution. Additional measurements to estimate acid dissociation constants would improve the manuscript.

Response 5: We thank the reviewer for this comment. We note that ADME and pharmacokinetic studies are beyond the scope of the current work. This work focussed on the ability of these compounds to chelate iron whilst acting as DAT substrates. However, such properties will indeed be studied when further investigations involving biodistribution aspects are undertaken.

Comment 6: A thorough investigation should be carried out to estimate the stability constants of the metal complexes and their speciation as a function of pH, so to predict the pFe3+ (-log[Fe3+free]) and compare the results with those of the leading compounds (deferiprone and Clioquinol). The Authors need to consider iron-hydroxo species that may form in solution as well. Furthermore, the Authors only report the data on compounds 1 and 6 (as supplementary), the main differences in absorption bands and at least a picture with Fe(III) complexes at the same molar ratio with all the ligands has to be added and commented on.

Response 6: This, again, is an insightful comment and such data will indeed be required when this work progresses to a more biological testing context. We highlight that in the current work, the isothermal calorimetry experiments were all carried out at a physiologically relevant pH of 7.4 (which was buffered – PBS). For completeness, we have included isothermal calorimetry data in the supplementary information for each of compounds.

Comment 7: EC50 (the antioxidant concentration necessary to decrease the initial amount of DPPH radical by 50%) could help compare the results. I am concerned about the methods the Authors used to assess antiradical activity. To my knowledge, in this assay, the absorbance is read at the maximum of DPPH (517 nm) and the decrease is used to define the EC50 as profusely reported (an example   https://doi.org/10.1111/cbdd.12847)

Response 7: We understand the concern about the choice of using a 520 nm filter instead of the 517nm filter (as indicated in the literature). While there is a slight difference (520 nm filter vs 517 nm filter) this should not pose a significant change to the measured activities as there is still significant absorption at this wavelength.

Comment 8: The Authors obtained the solid complexes and carried out some IR measurements. Are these data useful to the scope of the manuscript? Were the isolated complexes used to confirm the stoichiometry predicted in solution? If these results can provide some information, they should be properly added to the manuscript throughout the discussion.

Response 8: The IR data collected for the isolated were consistent with that expected but were not remarkable to any degree and so not discussed in the manuscript.

Comment 9: It would be interesting to test the isolated iron complexes for radical scavenging ability and compare them with the free ligands.

Response 9: We agree. This is a topic for investigation in future work.

Comment 10: The high affinity for the dopamine transporter could be on one side, as stated in the rationale of the manuscript, a good way of targeting the brain, but on the other side, it could be detrimental to iron complexation, being the transporter in competition to bind the chelator itself. The Authors should comment on this in the discussion.

Response 10: A good point. We have added comment on this topic to the conclusions section indicating that this should be considered in further work.

Reviewer 3 Report

Comments and Suggestions for Authors

This paper describes a well-designed medicinal chemistry study in which interesting structure/activity relationships were obtained for iron-chelating agents, with potential applications to the treatment of Parkinson's disease. The scientific material and the writing are very good overall, and I strongly recommend publishing this paper. I have some minor suggestions having to do with grammar and style, as follows:

Line 22: Replace "exhibiting" with "exhibit."

Line 28: Add a comma after "neuron damage."

Line 51: Add a comma after "dopaminergic cells and."

Line 52: Replace "it may be reasoned that" with "it is plausible that."

Line 64: Replace "deferriprone" with "deferiprone."

Lines 76-78: While this section mentions that the compounds can be protonated, and the X-ray structures are shown in the SI, I would explicitly state here that compounds 3 and 4 are crystallized as the dihydrochlorides, and that compound 5 is crystallized as the hydrochloride.

Schemes 1 and 2: This is up to the authors, but my preference in reaction schemes is to write reaction conditions over the reaction arrows, rather than in the caption.

Line 120: Add a comma after "MS ionization process."

Table 1: In the first line, the proper subscript should be added to "Ka" to make it "Ka." Also, I note that Kd is given in µM and Ka is given in M-1. I recommend either giving Kd in M or giving Ka in µM-1 for the sake of consistency.

Page 5, paragraph 1: I think the discussion of the DPPH assay would be enhanced by showing the structure of DPPH.

Line 215: Replace "cm-1" with "cm-1" (correct superscript).

References section: I noticed that some articles whose journals give article numbers, but not page numbers, do not list the article numbers, just the year and volume numbers and the DOI. Article numbers should be present along with DOIs.

Comments on the Quality of English Language

The quality of English is overall fine; I made some minor grammar and style suggestions in the comments.

Author Response

Comment 1: Line 22: Replace "exhibiting" with "exhibit."

Response 1: This has been done.

Comment 2: Line 28: Add a comma after "neuron damage."

Response 2: This has been done.

Comment 3: Line 51: Add a comma after "dopaminergic cells and."

Response 3: This has been done.

Comment 4: Line 52: Replace "it may be reasoned that" with "it is plausible that."

Response 4: A very nice improvement, this has been changed.

Comment 5: Line 64: Replace "deferriprone" with "deferiprone."

Response 5: This has been done.

Comment 6: Lines 76-78: While this section mentions that the compounds can be protonated, and the X-ray structures are shown in the SI, I would explicitly state here that compounds 3 and 4 are crystallized as the dihydrochlorides, and that compound 5 is crystallized as the hydrochloride.

Response 6: An excellent suggestion. The descriptive text has been added.

Comment 7: Schemes 1 and 2: This is up to the authors, but my preference in reaction schemes is to write reaction conditions over the reaction arrows, rather than in the caption.

Response 7: We would be happy to change the schemes if required. In the case of Schemes 1 and 2, moving the reaction conditions to above the arrows will require addition space for the schemes (roughly twice) as the scheme will break over two lines. We are happy to take the Editor’s advice.

Comment 8: Line 120: Add a comma after "MS ionization process."

Response 8: This has been done.

Comment 9: Table 1: In the first line, the proper subscript should be added to "Ka" to make it "Ka."

Response 9: This has been rectified.

Comment 10: Also, I note that Kd is given in µM and Ka is given in M-1. I recommend either giving Kd in M or giving Ka in µM-1 for the sake of consistency.

Response 10: We agree. Ka is now reported in units of µM-1.

Comment 11: Page 5, paragraph 1: I think the discussion of the DPPH assay would be enhanced by showing the structure of DPPH.

Response 11: An excellent suggestion. This has been added to the manuscript.

Comment 12: Line 215: Replace "cm-1" with "cm-1" (correct superscript).

Response 12: This has been rectified.

Comment 13: References section: I noticed that some articles whose journals give article numbers, but not page numbers, do not list the article numbers, just the year and volume numbers and the DOI. Article numbers should be present along with DOIs.

Response 13: These have been rectified.

Reviewer 4 Report

Comments and Suggestions for Authors

McDonagh group reported two series of compounds based on 8-hydroxyquinoline and deferiprone iron chelators that have amphetamine-like structures. The hydroxyquinoline-based compounds exhibiting stronger iron binding constants than those of the deferiprone derivatives in aqueous solution. Further more, molecular dynamics simulations showed that the hydroxyquinoline-based compounds generally bound well within the human dopamine transporter cavities. These compounds have been synthesized and characterized. The results demonstrate that these compounds are excellent candidates for diseases that are affected by iron-induced dopaminergic neuron damage. Over, this work can be accepted ASAP after revised the following concerns.

However, the key molecules should be provided the NMR spectrum copies. Some new method regarding the synthesis of quinoline related compounds can be referenced.

Author Response

Comment 1: … the key molecules should be provided the NMR spectrum copies.

Response 1: NMR spectra of the key compounds has been added as requested to the supplementary information.

Round 2

Reviewer 2 Report

Comments and Suggestions for Authors

Dear Authors,

you have addressed some of my concerns in your response, I still suggest reporting EC50 values for DPPH assay (not the data reported in the new table 2 that is redundant with fig. 3).

Author Response

Comment 1:  I still suggest reporting EC50 values for DPPH assay

Response: This has been included as suggested.